# Technological Sustainability and Artificial Intelligence Algor-ethics

**Alessandro Mantini** 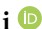

Faculty of Medicine and Surgery, Università Cattolica del Sacro Cuore, 00168 Rome, Italy; alessandro.mantini@unicatt.it

**Abstract:** Since 2018, a new terminology has been developed, called Algor-ethics, indicating the necessity for a dedicated study concerning the evaluation of an ethics applied to technology, to Algorithms and to Artificial Intelligence (AI). At the same time, since 1987, when the concept of sustainability was introduced, the discussion on this issue has become increasingly lively and has now spread to every area of life. In this paper, we would like to propose an application of the concept of sustainability to technological processes and in particular to the elaboration of AI systems. To reach this goal we will first try to build an ethical framework, here called Dynamical Techno-Algor-Ethical Composition, to define the interaction between the most important ethical ingredients involving the human person in relation to technology, taking a person-centered approach. Out of this will emerge a possible structure and definition of Technological Sustainability. The second step will consist of evaluating the process for the elaboration of an AI algorithm as a concrete application of the previously analyzed framework, to set ethical contents composing what we might call a good and sustainable algorithm.

**Keywords:** algor-ethics; technological sustainability; artificial intelligence; justice; power of service; freedom-creativity; anthropology; responsibility; eschatology; completion; pacification; meekness/minority; poor big data

## 1. Introduction

In this paper, we propose a study evaluating the possible basis of attempting the construction of an ethics that could confer meaning, motivations and impetus in using new technologies and Artificial Intelligence. A motivational and operational framework can in fact be created, for a wise interpretation of technological development, in order to orient it to the service of the human person and therefore to their good.

The post-modern era, with its increasingly pervasive technology, offers great benefits to humanity in terms of opportunities and resources to learn about and to improve quality of life. At the same time, precisely this technological development expresses the vitality, intelligence and creativity of mankind, as well as the deep bond that each generation has with the previous ones, from which it draws the wisdom, errors and knowledge to grow in the direction of improvement and progress.

$$From\ Uncertainty\ to\ Fear\ \bigvee\ From\ Uncertainty\ to\ Confidence \tag{1}$$

The new scientific paradigm, which places uncertainty as a new essential epistemological element, offers us a compelling and delicate challenge and dilemma, which we could summarize with this radical out/out:

In fact, trust presupposes a movement that pushes out of oneself, towards an otherness that invites us to progress in wisdom, and in this sense generates new creativity in turn, escaping from an emotional vision of the present that hides a fear of the future. At the same time, however, progress cannot unfold, or complete itself, if it lacks a fundamental ethical reflection or a wise moral reference that could move technological choices by operating a step by step enlightened discernment.

To set up an adequate ethical reflection, it is certainly necessary to take into account an international and intercultural dialogue, to converge in the search for a universal Algor-ethics, which can guide technological innovations towards the common good of humanity. In this paper, we then intend for Algor-ethics [1] to become a possible ethics concerning the development of technology and in particular of AI and related algorithms.

Sustainability is in turn a concept that was used for the first time in 1987 to try to orient development both in a synchronic sense, thus significantly linking today with the future, and in a diachronic way, simultaneously focusing on the different conditions of human needs and poverty, and introducing possible limits in order to respect environmental dynamism: "Sustainable development is development that meets the needs of the present without compromising the ability of future generations to meet their own needs. It contains within it two key concepts: the concept of 'needs', in particular the essential needs of the world's poor, to which overriding priority should be given; and the idea of limitations imposed by the state of technology and social organization on the environment's ability to meet present and future needs" [2].

We would therefore like to discuss a possible extension to Technological Sustainability in the context of an adequate ethical person-centered reference, and its application in the development of a consequent sustainable Algor-ethics applied to AI. In fact, putting the human person at the center of every technological project means providing primarily, from the initial stages of its development, for the common good of today and therefore of tomorrow, and at the same time it means promoting the human person so that they develop their skills in an interpersonal and relational horizon that builds the equality of dignity in the diversity of individuals. In this way, sustainability meets ethics, where the latter can offer concrete and well-founded coordinates, first of value and then operational, for a technological development that respects humanity, allowing it to grow towards fullness.

## 2. Technology and Anthropology

In this section, we intend to put in evidence, as a starting reference, the necessary interconnection, to be discovered and preserved, between technology and anthropology, in order to maintain the link with the human person while building technology itself. This will be the first step for a sustainable Algor-ethics.

A governance of technologies has been primarily identified with the need to give value and centrality to the human person, because through the questions arising from the increasingly pervasive interaction of technology with individuals, which generates a technological and evolutionary uncertainty, AI (and technology in general) would be legitimate "to the extent that allowed men to remain equal to each other and favored the inalienable desire of each to seek their own happiness" [3], that is, the fullness of life. A second contribution is given by social relations because "the institutional world, the academic world and technology companies should reflect together to implement a Governance" in the horizon of the equality of human dignity, for "authentic human development" at the level of individuals and entire societies in which the person is always the goal [3]. Finally, a third element is the crucial link between technology and its reference to anthropology, bearing in mind that "the technological artifact has no ethical dimension since its only purpose is efficiency and it is the will of man that determines its correct use"; in fact, technological intentionality requires freedom and responsibility, which are closely related to the culture in which they are inserted, with its values, its morality and its ethos (this is the so-called technological multistability) [3].

On the other hand, it is necessary to recognize the original value of technology, in order to develop an adequate ethical reflection that does not get lost in prejudices, which close the mind, or in the uncontrolled permissiveness, which justifies everything. In fact, technology has been defined as a sort of window "that allows us to perceive a particular dimension of the human being" [4].

It is not possible, honestly, to consider man and his integral condition of personal being, in his physical, intellectual, spiritual and conscientious dimensions, without referring to a

Creator God, or at least to a question about Him; in fact, the basic condition of man is that of having received himself and not that of having made himself. Similarly, technology is not possible without its author, who is the Person, and without a well-founded anthropology.

Starting from man's intrinsic relationality and expressiveness with respect to the world around him, precious categories emerge while we try to highlight some distinctive traits of man himself, according to the proposals of Paolo Benanti that we will now report and analyze [4]: first, technique can be considered as a 'real symbol', because the very relationality of man, in addition to the body, which is the first essential mediator as it characterizes his identity in the unified unity of the person, is also realized through technological artifacts. They are in fact "representative signifiers of the spiritual self of the person who has in his essence a being called to exist-with; . . . a technical artifact is a real symbol of the person and of his actions". Technique is also an 'oriented history' because man interacts with the world attracted by its mystery and, through his cognitive dynamism, modifies it by entering a relationship and consequently builds a story in which "the artifact exists and stands in the original space of relationship that man is". In this sense, then "the technological artifact participates, mediates and is a container of that dimension of historicity and worldliness that characterizes human experience; . . . the person, constantly turned to God, absolute mystery, participates, realizes and verifies this orientation also through worldly experience mediated by technological artifacts". Technique is also a place of 'freedom', because man receives this gift and is called to respond responsibly to God's calling through his choices. While technique speaks about man in his historical and transcendent being, through his mediation, it "is the place where the opacity of the objectification of transcendental freedom is perceived more clearly". Therefore, man's freedom and its 'drama' are expressed around the artifacts. "The finality of freedom is experienced and implemented mainly in the technique: every single decision and technical operation irreversibly changes the existing, transforming it, making it impossible to return to the previous conditions". All these aspects, as can be seen, contribute to building a bridge with the concept of sustainability, in its reference to reality, history and the transformation of the existing, in close connection with technology. Finally, technique is an experience of 'creaturality' as the original experience of the human being who discovers that he is not the creator of himself, and who, thanks to the experience of being able to 'do' and 'create', questions him further, first about himself and then on the meaning of his action. Therefore, "technique, as an authentic and profound experience of freedom and responsibility in the world and in history, is then an authentic experience of creaturality; . . . technology is also the place of man's profound autonomy". This ambivalence of the technique between creaturality, that means vital dependence, and autonomy continually stimulates, on the one hand the search for truth, and on the other the foundation of an ethics that illuminates its dynamism.

The imperative that is clearly outlined is thus to consider technology not separated from a renewed, extended and necessary reflection about the human person: anthropology and technology are in fact much more interconnected than one might think at first glance:

$$\text{Anthropology} \rightleftarrows \text{Technology} \tag{2}$$

The alternative is a reductionism, which is moreover in line with the typical and characteristic tendency of the post-modern condition, which loses sight of the dynamism of complexity, of the multi-level structuring of reality, of the inevitable interconnection of diversities, of the synergic knowledge and of the need for the expansion of rationality. Post-modernity seems, in fact, oriented towards an epistemological reductionism, fragmenting knowledge and dividing the disciplines, isolating and making them autonomous, and towards a methodological reductionism, as it concentrates the approach to reality by enhancing the mono-dimensionality of empirical science alone. This reductionism also leads to an unbalancing of the strength of the bi-directionality of the relationship indicated above, between anthropology and technology, weakening its links.

In this way the ethical problem, before being referred to the technology itself, must deal with the recovery of the solidity of the aforementioned bonds, to read the technological challenge in its integral dimension, in which the human person is the driving force:

$$Tecnhology \leftrightarrow Person\ centered \tag{3}$$

### 3. Social and Transcendental Coordinates of Technology

Within the above context, in which anthropology assumes a fundamental role for technological development, in order to have a clear person-centered approach, in this section we intend to highlight those pillars that we consider in a direct and structural relation with technology as it is emerging today, and is therefore fundamental in our proposal. We thus divide them into two dimensions: the horizontal ethical coordinates: responsibility, justice and power, that define the Social Ethics of Technology; and the vertical pillars, that represent a sort of fermentation and dynamic development ground, such as: anthropology, freedom and creativity and finally teleology and eschatology, collected as the Transcendental Motives for AI and Technology [5]. The latter inscribe, in our context, an orientation of meaning, giving the necessary breath to the Social Ethics of Technology, which otherwise would remain prisoners of a freedom without a source. These coordinates are not static but are strictly interconnected in the dynamism of human life so that the ethical attitude could emerge as a result of their synergy. For this reason, we first need to basically describe these horizontal and vertical ethical pillars, first starting from the Social Ethics of Technology.

Responsibility can be considered as a consequence emerging from five verbs, defining the attitude of mankind toward the Cosmos: the first is 'to Dominate', that is linked to the concept of Lordship, not as a despotic dominion, but as a wise regulation, like a 'dominus', enlightened and benevolent called to responsibility; the second is 'to Subjugate', that expresses the function of control, in particular referring to the Earth, therefore to each of its fruits and to each of its products, cultivable and extractable, as resources for human development, sustainable and integral; the third is 'to Serve', as an attitude and a disposal of service with respect to a gift received, that means a service in the horizon of the relationship; the fourth is 'to Guard', that can be understood as 'watching' with attention and respect, in the sense of taking care of, and as 'preserving'. This is in reference to the development of a fine and conscious intelligence, to learn the mechanisms, to know the beauty and the preciousness, but also to discover a wise use and a possible transformation, oriented to the good of the Cosmos. This custody is therefore not something static, but it refers to the potentialities and talents belonging to each human person and to their relational interactions, in order to promote the common good. The last verb is 'to Call', to indicate both calling by name and proclaiming. It fully values responsibility and dignity but also trust. In fact, giving a name implies knowledge and knowledge presupposes a careful and intelligent commitment, possible only in the context of a given and precious dignity. Man in fact can surprisingly 'understand' the Cosmos and can transform and use it, within the horizon of the verbal categories just listed, in order never to stop recognizing and listening carefully; the continuous calling it is addressing to him also through the Beauty of creation itself: "The very fact that the totality of our sense experiences is such that by means of thinking (operations with concepts, and the creation and use of definite functional relations between them, and the coordination of sense experiences to these concepts) it can be put in order, this fact is one which leaves us in awe, but which we shall never understand. One may say 'the eternal mystery of the world is its comprehensibility'. In speaking here of 'comprehensibility', the expression is used in its most modest sense. It implies: the production of some sort of order among sense impressions, this order being produced by the creation of general concepts, relations between these concepts, and by definite relations of some kind between the concepts and sense experience. It is in this sense that the world of our sense experiences is comprehensible. The fact that it is comprehensible is a miracle" [6].

Responsibility is thus, in our point of view, a direct consequence of the person-centered approach and emphasizes, first of all, the recognition of the dimension of the 'calling', to which it represents precisely a response. This 'calling' resumes and completes the sense of all the other meanings just explored in relation to responsibility. The latter does not arise then in itself (as post-modernity seems to support, whereby it becomes ephemeral and changing depending on the subjective emotions), but as a consequence of a 'calling' that can come from multiple interconnected levels. Every possibility, which emerges in the panorama of history and progress, in fact poses a question to human beings and their consciences, which consequently need to be trained to recognize their solicitations, to be answered with ethical choices. In addition, the relationships in which man is involved stimulate him to respond ethically and responsibly. We therefore have a fourfold level of responsibility, which shows the flowering of this ethical pillar in the vitality of the human person:

$$Responsibility = Calling \Rightarrow Answer \rightarrow \begin{cases} to\ God \\ to\ Knowledge \\ to\ History \\ to\ Relations \end{cases} \tag{4}$$

The second step is the orientation towards Justice. This is a very important issue, primarily considering justice to people as persons [7]. In this sense, the reference both to the good of the single person and to the common good of society and humanity is immediate:

$$Justice \Rightarrow Good \rightarrow \begin{cases} of\ the\ Person \\ Common \end{cases} \tag{5}$$

Justice is the way through which the ethical commitment of charity takes substance, which is in turn embodied precisely in Justice, without, however, identifying itself with it; in fact: "Charity . . . has its first expression in Justice, which . . . informed by charity, participates in the theological and salvific tension" [8]. It is in this sense that charity prevents justice from objectifying itself, that is, from remaining static and arid, just as, without justice, charity would be abstract and unproductive: "Justice gives charity a realistic incarnational character" [8].

Finally, we have the question of Power, which needs to be 'dimensioned' in an ethical sense. This is a particularly critical experience for humans, also in reference to technology and its use, due to their fragility. Today the technique is recognized, for its pervasiveness, as a real 'method', an 'environment' and a 'way of life' [5], as well as a sort of 'structure (pattern) of reality', or better still a real 'key to modernity' [9].

The problem of power emerges in the relationship between technology, man and society, and it can be said that "Technology is Power: technology is a powerful and determining factor" [5] and that "Technique is a realization, a fulfillment and an increase in the spirit of power that leads to a polarization of man on power" [9].

Power must therefore be analyzed carefully, as, on the one hand, it has to do with a sort of 'form' of the contemporary world, due to the involvement of man, who uses it more and more; on the other hand, man entrusts himself to it by delegating the custody of his own history, if not even the 'guide' of it. The question of power linked to technology is then connected with the dimension of domination, and involves man who, in his weakness, is called in turn to recognize the preciousness of his dignity and his potentialities, which are precisely given. These latter two polarities (weakness and potentiality) are to be inscribed in the context of an order to be discovered rather than defined: this is the order of the Cosmos, which already contains in itself every stimulus for the realization, fulfilment and increasing

of man's dignity along the search for the truth. In this context, power should be experienced within the unitary dynamism that involves the Cosmos, the person and society:

$$Power \Rightarrow Confidence \rightarrow \begin{cases} Cosmos \\ Human\ Person \\ Society \end{cases} \tag{6}$$

After the horizontal coordinates for a Social Ethics of Technology, the vertical pillars are the Transcendental Motives for Technology. These can be considered as three dynamic 'engines' that could orient technology itself in a fermentation process that on one side evaluates human's potentialities and on the other maintains a look to human dignity and its fulfillment.

Starting with the lower level, we thus have Anthropology, which characterizes the preciousness of man as an integral person: "we speak of man as a person, and we mean both his endowment with the spirit, which goes beyond materiality and physicality, and the independence and freedom that is founded in it" [5].

Going up we then meet the level of Freedom and Creativity, which shine as gifts and opportunities received and given, and only in this context they both can be expressed to make the person flourish toward the good, according to the logic of an ordered complexity, in a historical horizon rich in meaning, because it is not self-made.

Finally, the decisive level is the Teleological or rather the Eschatological level, which supports the whole system because it enlarges the horizon and keeps the 'skies' open, that is, it enlightens the real possibility of a significant, substantial and oriented progress: it "is about the completion of the individual as well as the completion of the whole creation" [5]. This orientation element confers the dimension of finality onto the human activity, deriving from the intrinsic value of the human person, which is therefore called not only to be, but also to flourish.

We can then summarize by saying that: "The technological artifact and consequently the world of technique/technology, is the epiphenomenon of its [man's] essence, the existential place, the techno-human condition precisely in which greatness is shown of its vocation in the fragility of its constitution" [4].

## 4. Technological Sustainability: The Dynamical Techno-Algor-ethical Composition

Receiving as inputs the social ethics of technology (responsibility, justice and power) on the one hand and on the other the transcendental motives for technology (anthropology, freedom-creativity and eschatology), we now reach the core of this paper, proposing a specific composition in order to appropriately structure them in a dynamic, operational and propulsive relationship: dynamic, because all the individual elements of such a system should be continually interconnected in an active dialogue, capable of wisely facing the complexity of reality and the challenges it poses; operational, as the integral relationships between the components should be able to descend to the level of implementation, and therefore have strong roots and luminous foundations; propulsive, because an organic reading, such as the one we are proposing, in addition to offering a solid elasticity, should finally be able to push towards a fullness of meaning and choices regarding the good of the human person.

With these premises, we now come to introduce two elements that we believe are decisive for guaranteeing the right glue and the necessary elasticity to all the analyzed elements, the horizontal and the vertical ones, in order to structure bonds with the characteristics described above. Humanity, in fact, is involved, and in a certain sense also expresses itself in technology; therefore, the ethical coordinates need to interface with the multi-dimensionality of human life, which overcomes determinism. At the same time, as we have seen, the fundamental foundations that support and direct the ethics of technology, making it effective through the freedom and creativity of the human person in his personal and community action (in the sense also of the international scientific community), require a dynamic and vital qualification. We believe we can identify this multi-dimensionality and

qualification in a potentially extremely effective attitude called Service of Love or Power of Service (first element), which is able to harmonize the structure we propose, precisely in the horizon of meaning that calls the human person to his Fullness (second element). We will then obtain what we call the Dynamical Techno-Algor-Ethical Composition, which constitutes the ethical environment in which to develop specific traits and operational provisions in the individual technological and/or algorithmic design applications (Figure 1).

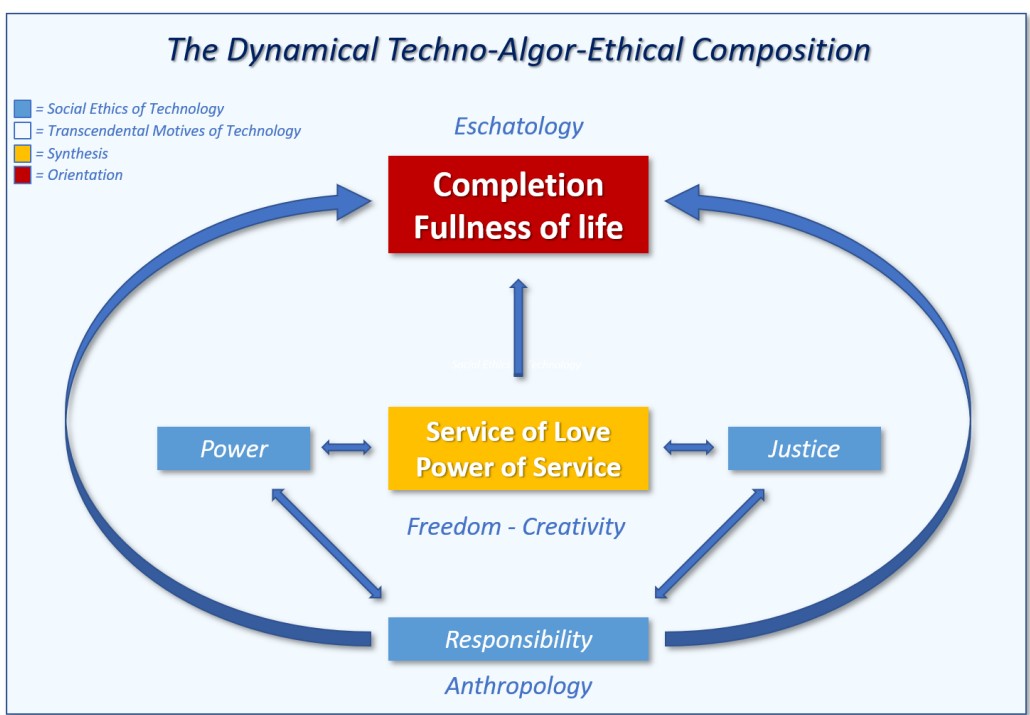

**Figure 1.** The Dynamical Techno-Algor-Ethical Composition.

The Service of Love or Power of Service, is a decisive element towards which, on the one hand, the forces of power and justice should converge in order to be enriched in meaning with respect to the human person, which represents the center, and on the other hand, freedom and creativity could be significantly expressed within this context in which Love is manifested as the most precious gift received that asks to be donated (service). Power needs to be anchored on an original value (love) and cannot be left alone at the mercy of itself; but, at the same time, it needs a specification (service) that could qualify its horizon. In the same way, justice must be placed in a position in which could orient itself (service) and also be founded (love). The result is the two-way link that power and justice have with responsibility, which in turn is solidly built on a consistent anthropology. Responsibility, gradually nourished in the discernment of a formed and listening conscience, thus allows the integral person to learn and to manage power and justice in a wise way. Power therefore becomes service (having in mind that 'to serve is to reign'), and the justice that derives from it is a constitutive existential/relational condition and an opportunity for continuous progress and creative flowering to be expressed. Justice, in fact, if it is oriented towards service and love, rediscovers its roots. The Service of Love and the Power of Service thus offer a clear identity to the expression of freedom and creativity, typical of the technological innovation, and they become possible, abundant, luminous and capable of fully and reasonably renewing themselves in the horizon of the truth that is now colored with charity. The Service of Love assumes the role of 'semantic and pulsating glue', because it confers meaning and vital order, with an almost cardiac rhythm, onto the three pillars of the social ethics of technology. These coordinates are expanded on two levels of the transcendental motives: the anthropological one, which forms the basis for responsibility, and the freedom-creativity one, which expands its operations. Justice, power

and responsibility are nourished, therefore, by dynamic relationships, which indicate their interdependence, and above all the vital link with the dimension of the Service of Love, moving in a real virtuous and multi-dimensional hermeneutic circle. The latter expresses itself in a systolic beat, concerning the progressive discernment and research for an ethical value (Figure 2a), and in a diastolic beat, which allows its colorful and lively expression precisely in the ever-new context of a person-centered technology (Figure 2b). The Service of Love is therefore the element of synthesis of our composition.

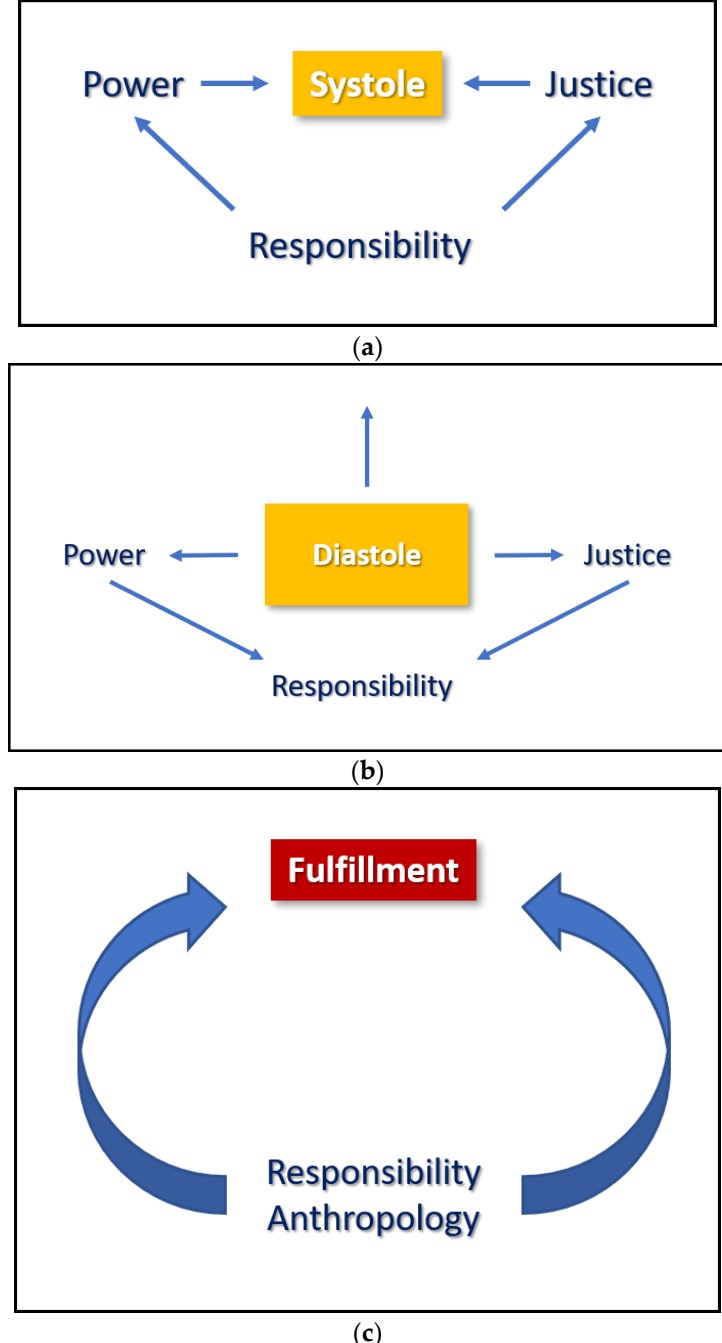

**Figure 2.** (**a**). Systolic beat of Service of Love. (**b**). Diastolic beat of Service of Love. (**c**). Breath of Responsibility.

The orientation is represented by the *Fulfillment* and *Fullness of Life*, because this first virtuous and multi-dimensional hermeneutic circle is constantly projecting and orienting itself towards completion, at each 'beat' of the Service of Love, according to the structural

calling we have highlighted above. The eschatological dimension in fact confers the consistency of the end (in the sense of finality not of finiteness), to which every human dynamism tends: precisely the fullness of life. In a person-centered perspective, technology should therefore receive and participate in this intimate human movement because it has to be rediscovered as 'person moved'. Not only the beats of the virtuous multi-dimensional hermeneutic circle, supplied by the Service of Love, but also the breath of responsibility, must point to completion. In this case we no longer have a 'cardiac dynamism', i.e., systolic and diastolic, but a constant flow, as a tension of convergence, a continuously oriented passage to the limit, finely tuned with ever greater precision and awareness. The fulfillment is thus the element of orientation in our composition (Figure 2c).

In this way the Social Ethics of Technology and the Transcendental Motives of Technology are dynamically connected and receive, as a living and enlarged heart, the Service of Love, which constitutes their 'beating' synthetic core, and also the completion, which represents in turn their propulsive orientation term: all this defines the Dynamical Techno-Algor-Ethical Composition or Ethical Framework for technology finally composed with all these macro-ingredients:

$$Social\ Ethics\ of\ Technology \rightarrow \begin{cases} Responsibility \\ Justice \\ Power \end{cases} \tag{7}$$

$$Transcendental\ Motives\ for\ Technology \rightarrow \begin{cases} Anthropology \\ Freedom-Creativity \\ Eschatology \end{cases} \tag{8}$$

$$Multi-Dimensional\ Dynamism \rightarrow \begin{cases} Service\ of\ Love \\ Fulfillment \end{cases} \tag{9}$$

This dynamic composition acts as an ethical reference for technology and AI, which should be inserted within it, and in this way could lead to the construction of a sustainable technology and Algor-ethics.

At this point we can summarize technological sustainability, offering this definition: "technological sustainability is the opportunity to develop an authentic human progress according to an ethically consistent interaction with the technique, considered as a person-centered dynamical service. Technology, in this context, assumes its own role to sustain mankind's freedom and creativity in order to respect human dignity and to facilitate, embracing with responsibility all humanity in the service of love, the expression of its common good in a diachronic path toward fulfillment".

## 5. A Possible Algor-ethics Structure of AI Ant Its Sustainability

In this section we propose a specific application of the ethical framework that we have presented, to Artificial Intelligence. The Dynamical Techno-Algor-Ethical Composition could be in fact, in our proposal, the framework in which a researcher may move to build and structure a sustainable technology or algorithm. In the following application to AI, we will show how we can obtain specific features in each individual design phase, to define a Good Algorithm, emerging from the inside of the previously discussed framework.

In particular, it is a question of entering into each of the individual development steps of an Artificial Intelligence Algorithm by making conscious ethical choices that on the one hand have the objective of defining the process in a sustainable way, and on the other hand dispose the human person, who carries out and realizes the project, in an ethically consistent perspective. The result will therefore be to define an AI Sustainable Algor-ethics both in the design and in the use phases, structured in the wake of the Dynamical Techno-Algor-Ethical Composition, and therefore starting from the person-centered perspective. For this purpose, we choose to split an AI project into 5 subsystems, following the different steps that realize its development:

1. Data: this is the initial subsystem, which refers to the data to be used, along the different phases of choice and evaluation of the source, collection, selection and preliminary analysis. It refers to the type of data to work with, specifically those used as input and as output, considered here for a qualitative and quantitative evaluation;
2. Model: subsystem relating to the choice and implementation of the data analysis model;
3. Learning: subsystem that deals with the choice and use of the automatic learning methodology;
4. Validation: subsystem that concerns the validation phase of the process from an internal and external point of view to ensure its reliability;
5. Reporting–Presentation: conclusive subsystem that concerns the presentation of the results and their relevance, that is the final user interface.

From these subsystems we now suggest the construction, step by step, of an ethical structure that allows us to proceed with the qualification of a 'Good Algorithm' of AI, in order for them to become Algor-ethical subsystems.

In fact, we believe that an Algor-ethical monitoring should not be developed simply starting from the inputs and outputs, but should proceed by analyzing all the phases of the algorithmic process, thus entering the technological and algorithmic box to evaluate all the elaboration chain from the inside.

Analyzing the contents of the processing steps, we can build an ethics for the AI project that in turn concerns man and the way he manages technology and the AI itself, but at the same time defines the quality of the algorithm and its use. If it is true that no technological artifact is ever neutral in itself, an Algor-ethics must somehow both color the technology of the good and enlighten mankind in his choices towards the good. The aim of each following paragraph is to answer to the question: "which ethical trait could emerge from each single algorithmic step drawing on from the Dynamical Techno-Algor-Ethical Composition?".

### 5.1. Data Ethics: Poverty of Spirit

The data represent a numerical sampling, therefore a translation into digital numbers of quantities associated with the dynamic and continuous life of a person, or with natural phenomena.

The widespread use of digitization highlights an ethical urgency relating not only to data processing, but also to their custody (who keeps them, how they keep them and how they use them) and in particular their meaning. Therefore, managing Big Data requires particular ethical attention, since it is the very first step that collects a sort of digitalization of man and reality. It is a form of sampling to provide an adequate and sufficient virtual image, which will then be artificially processed or elaborated.

It therefore seems appropriate to combine this precious moment of compiling and using an algorithm with the attitude of the 'poverty of spirit'. This provision, on the one hand, should always be maintained throughout the evolution of the technological process (design, construction and use); on the other hand it allows us to interface with Big Data in an ethically sustainable way. In our context the 'poverty of spirit' consists in fact of a disposal that guides us to consider the data for what they really are, 'respecting them' within the poverty/totality polarity. In fact, Big Data express a radical and unsurpassable poverty, due to the very nature of the data themselves, which consequently calls man to a recognition that exceeds and goes beyond the enthusiasm aroused by their use and the results that can be obtained from them.

Although they may be very accurate, they are, however, an expression of a poverty of content compared to reality, a poverty certainly useful and usable also for great results, but this poverty represents an 'existential datum' of the Big Data, which we then could rename as 'Poor Big Data'. Quantity should not be in fact confused with quality, to which this poverty of data always and in any case refers, as their constitutive characteristic that cannot be overcome. Recognizing the structural poverty of data is not a mere ethical exercise, but disposes an operational attitude that can qualify not only man in the approach to technology, but also technology itself, that could, in this way, increasingly be valued

because it is inserted in its own horizon as a work of human hands and dedicated to his service (this is the diaconal dimension of technology).

'Respect the data' means not to give them a disproportionate power and therefore evaluate the human person in his dignity, to enhance reality in its multi-level depth and to enhance the technology in its service.

Poor Big Data, therefore, once recognized as such, can be correctly contextualized as part of a totality. They certainly represent the main 'meal' for digital technology, but considering reality in a multi-level perspective, even technology cannot presume to travel alone, but must be placed in a wider context. We will then speak of 'technology in context' to indicate a technology emerging from the human person, from his dignity and creativity and from his intelligence and spirituality; a technology emerging from reality, using its materials and enriching (sometimes destroying) it with new syntheses; a non-living technology, which emerges instead in a living, dynamic, continuous, contingent and unpredictable context.

Poor Big Data are thus part of a dynamic and profound 'totality', of which they represent a very small and static fraction, as a partial perspective and a mono-level reduction.

It is about a poverty in a wholeness, which helps us to integrate, contextualizing them, all parts and all levels, in a single greater reality that always surpasses us for its complex and deeply intelligible characterization. The data collected, even when they become Big Data, are therefore still Poor Big Data, statically contained in the dynamic complexity of the real, of which they in fact represent a pale reduction. Here is the first step for a sustainable ethical AI.

### 5.2. Model Ethics: Meekness and Minority

Starting from Poor Big Data, the researcher then moves on to design mathematical models that can best exploit the properties and information content, in order to optimize their reading. The model is a synthetic and artificial representation of a stretch of reality, that is intended to be reproduced in order to develop simulations, forecasts or imitations for various needs. But the model is also a necessary and appropriately chosen mathematical algorithm to process data efficiently.

Designing a model means first of all learning about the reality to be modelled, extrapolating its contents and salient aspects, its facets and useful perspectives, as well as the properties of interest for the purpose for which it is being studied, in order to select appropriate simulation techniques, emulation or processing.

The potential of models, which can be linear, non-linear or even neural, are evident today in many sectors of technology and the ordinary life of men in post-modern society.

Faced with the importance and diffusion of models, we then propose to combine the attitude of 'meekness' at this stage of technological development.

Meekness could be the disposition with which man builds the model, freeing it from the presumption of being in the likeness of man. At the same time, man could confer to the model the traits of meekness, as long as it remains in the 'minority' with respect to the reality.

The 'meekness of the model' is then its 'technological minority', a minority that does not mortify its power and subtlety, but which instead gives it back its strength at least in two directions: that of its increasingly advanced potential and that of the reference and return to man not only as an author, but also as a decision maker, evaluator and measurer.

The meekness and technological minority of the model, and in general of each technical product, are characteristics that emerge from technology, thus offering it a double impetus:

$$Earth \leftarrow \underbrace{Technological\ Minority}_{Meekness} \rightarrow Mankind \qquad (10)$$

Towards humanity because technology continually depends on it, and without the human person it can neither be conceived nor used; towards the Earth (and the Cosmos) because it is an expression of it, but without the breath of life, and participating to the progress of man called to use the Earth on his way toward fullness.

### 5.3. Learning Ethics: Hunger and Thirst for the Truth

Alongside the development of the models necessary to process the data as completely as possible, to extrapolate the information content useful for the needs for which they were collected, perhaps the most fascinating step is to train the machine through that data.

Machine Learning consists precisely in using data in at least three distinct and related phases: first to train the machine, that is the first real 'learning' step, aimed at defining the parameters inside the model, so that it learns how to become familiar with that type of data (which are the training data); this is the learning process in which the machine somehow adapts to recognize the structure of the data it receives, in order to then be able to recognize, at the end of the process, similar data with a certain flexibility and precision. A second step is to test the machine, that consists in using part of the data collected (test data) to verify, at the end of the learning phase, that the machine has actually learned from the training data, showing that it can effectively recognize them with a tolerable margin of error. The last phase is to use the machine, when the model is now subjected to new data, those of ordinary use, to classify them based on what it has learned.

This is a real process of artificial learning, which therefore simulates natural processes, even though it inevitably lacks many uniquely recognizable passages with respect to human learning.

Machine Learning defines and tunes numerical parameters, structured in a more or less complex network of interconnections (linear, non-linear, two- or three- or multi-dimensional, neural...), statically (parameters established during the training phase) or dynamically (parameters established during the training phase and then gradually refined with use). However, these are numerical parameters which, as they are derived in an algorithmic way from Poor Big Data, in turn should be considered in the same way as Poor Parameters. The sequence is therefore as follows:

$$Poor\ Big\ Data \rightarrow Model \rightarrow Poor\ Parameters \rightarrow Learning \tag{11}$$

These parameters, which define an artificial learning model, always carry with them a more or less evident component of error, partly due to the quality of the data themselves and partly to the inevitable limitations of the model. This error can be refined through various mechanisms, but it always remains an integral part of the automatic process. It also lacks, with respect to human learning, the dimensions of relationality and creativity, of conscience, emotions, feelings, judgments and previous lived experience, freedom, neural conformation and the uniqueness of the Person, in his unrepeatable unity. So, we could summarize saying: "when faced with Machine Learning, 'remember to learn'". We certainly admire AI and Machine Learning for the results they allow us to obtain and, above all, for the technological and creative level reached by mankind, but all this offers and should continue to offer the researcher and everyone the opportunity to feel 'hungry and thirsty' for human learning and therefore for knowledge and truth.

An ethic of artificial learning must thus be based, in our opinion, on the appeal to hunger and thirst for knowledge that it offers to human learning. In fact, in the case of technology, knowledge tents to be always and only technical, reaching further data, while in the human case knowledge is always oriented towards the relationship. This is then the hunger and thirst that respect the dignity of man, who should remember to learn:

$$Artificial\ Learning \rightarrow Data \tag{12}$$

$$Human\ Learning \rightarrow Relation \rightarrow Person \tag{13}$$

### 5.4. Validation Ethics: Pacification

Once the results of the data processing and learning parameters have been obtained, the precious problem of their validation arises, which refers to the verification of their reliability for the operating phase. It is a question of evaluating the quality of the results, with additional appropriate parameters, both inside and outside the system.

In fact, the results will then be applied to the different contexts for which they were developed (technical, commercial, medical, . . . ), and depending on the sensitivity of the sector, they will require particularly precise degrees of verification, i.e., determining reliability parameters, which will once again be given summarizing the margin of error of the result with respect to reality.

In the validation of the results, therefore, the evaluation of the error intervenes in a decisive way to express the weight of certainty: it is in fact an indirect process.

In this part of the journey in the technological algorithm, we recognize the greater need for 'peace' between the various competing fronts: the expectations of the designers; the quality of the input data; the quality of the model; the error of the results (standard deviation); the expectations and satisfaction of users.

This first type of 'peace operation' can be defined as fine tuning; it is in fact a question of tuning different parameters, different processing qualities and different weights that can also conflict with each other, requiring design compromises. The latter are projected onto the final error and highlight a characteristic of the data and artificial processing, which is that of sampling the reality in a static way, eliminating the essential dimension of its dynamic complexity. This happens for reasons intrinsic to what is artificial, even in multi-dimensional and adaptive modelling.

We then see the validation of the data as a real peace-making process, where the most essential pacification is that between what is typical of human evaluation and what is numerical evaluation. In fact, human evaluation contains elements which, even if they may not always lead to absolute certainty, refer to an integral experience that cannot be reproduced and cannot be codified.

The cognitive process of reality, and therefore that for the evaluation of truth, represents a determining and characterizing element for humans, as it is animated by the peculiar faculties which are not communicable to machines. They are in fact on a different ontological level. Furthermore, the human person does not have the 'credentials' to be able to communicate and give these faculties, which are intelligence, creativity, judgment, conscience, morality, etc., while he is only an amazed user, called to make them bear fruit.

The pacification we are talking about therefore refers precisely to this ethical dimension of evaluation, which cannot in any way be attributed to the machine and algorithm, which are lacking of the essential components of the evaluating process. In the artificial chain of evaluation there is no imagination, intellectual intuition or judgment, which in the artificial does not have a predicative function (as for the Human Person) but of classification. It does not in fact reach being; it does not reach the proper form, but it reaches a statistical, numerical, qualitative, distributive, organizational and precisely classificative content. It is a more or less complex manipulation of data, not a knowledge or a judgment. The evaluation of error or certainty, in the artificial case, becomes a form of comparison between data and on the data (correlation). Certainty, on the other hand, belongs to the phase of judgment understood as an "act of the intellect with which the 'adaequatio of the intellect itself' is expressed with the things known as they are known" [10]. In the machine there is no exit toward a subject, but the evaluation mechanism resides entirely within the system and the algorithm, devoid of any metaphysical dimension. This means that an algorithm processes, manages and evaluates data, obtaining further data and, in this way, it starts and reaches the object level only, remaining one-dimensional and non-relational. On the other hand, a human evaluation starts from the data, considered precisely as an object, and the cognitive process takes place as the data 'go out' towards the human subject, opening the process to multidimensionality. This ethical pacification is then required precisely by the clear ontological difference between Homo Sapiens and Machine:

$$Algorithmic\ Evaluation \mapsto Data - Data \Leftrightarrow Object - Object \tag{14}$$

$$Human\ Evaluation \mapsto Data - Mankind \xleftrightarrow[Pacification]{} Object - Subject \tag{15}$$

The first 'pacification' consists therefore of not speaking of Artificial Intelligence but of Artificial Processing or Elaboration, and the second in not referring to Machine Sapiens but always to Homo Sapiens also if enhanced with the aid of artificial instruments. The machine can never be wise, while the human person partakes of a personal wisdom. The synergistic and ethically balanced complex of human and artificial can instead be wisely considered by virtue of the human component alone:

$$\underbrace{Artificial\ Intelligence}_{Not\ Pacified} \mapsto \underbrace{Artificial\ Elaboration}_{Pacified} \tag{16}$$

$$\underbrace{Machine\ Sapiens}_{Not\ Pacified} \mapsto \underbrace{(Human - Artificial)\ Sapiens}_{Pacificed} \tag{17}$$

Pacifying the moment of evaluation is therefore an essential step, because it places the truth of relationships and differences, restoring the dignity of knowledge to the human person. From this pacification derives the ethical evaluation of the interaction (not relationship!) with the machine and with the results it can produce.

*5.5. Presentation Ethics: Purity of Heart*

The final dimension of our algorithmic process consists in the phase of data presentation, that is, the interface with the user. It is a very important system as it involves the dialogic-communicative aspect. Through the communication process, in fact, a whole series of multidimensional inner reactions are generated in the human person that contribute to his wealth of experience, from which he can draw on to choose and grow. The languages of images and voice/music are always deeply evocative as they are capable of fixing the memory, not only of time and space but also of interiority.

The moment of data presentation, and therefore of the construction of the user interface, needs a careful and precious ethical evaluation. In fact, its evocative 'potentials' are capable of influencing the human person in his choices and in his inner, sentimental and emotional dynamisms, which then naturally have repercussions in the existential and spiritual dimension.

We can then schematize an internal dynamism of 'presenting', which in itself indicates an ethical trace, because it refers to imaging (to see) and speech (to say), as follows:

$$Presentation \rightarrow \left\{ \begin{array}{l} To\ say \\ To\ see \end{array} \right. \rightarrow Relation \rightarrow Goodness \tag{18}$$

Each presentation then generates a reaction, that leads to an answer, which in turn draws on a story, not simply on data, but on a life, not simply on a statistic:

$$\underbrace{Presentation}_{Data+Interface} \mapsto \underbrace{Answer}_{Person+Life} \tag{19}$$

There is therefore a disproportion, or rather a cause-effect incommensurability, and here lies the heart of the ethical discourse concerning this last step: data and interface have the strength to act on the person and on life, especially in the health sector, when AI is used as a tool for improving care, but in general the impact of 'presentation', in this digital age has strong resonances on the inside of the persons. In fact, very deep chords of the human person are touched, precisely in his interiority and spirituality, and it is therefore important that the Algor-ethics appeals to a 'pure' disposition of the person who designs, and that defines the 'dialogic' dimension man-machine. This could be a fruitful and sincere way to present a great expression of human intelligence, which is technology, by placing it in its appropriate context of service to humanity and also of strengthening its faculties.

The results presented with the attitude of the 'pure of heart' will then offer the user the opportunity to see the person and not just the machine, and even himself, perceiving

the expansion to the mystery that, in the humble and non-imposing attitude of technology, directs him to open the doors of his interiority and spirituality, to become more and more himself, not remaining just in the technological level.

In this way, technology could really serve the integral human person and enhance his faculties, allowing the gifts that characterize him to dare where humanity alone could not reach.

*5.6. Discussion*

What we have tried to define is a sort of ascent towards a mountain, which certainly costs effort, probably also in the face of the novelty of the topics covered, but certainly, once the summit is reached, it allows us to enjoy an organic panorama. It is also possible to perceive the perspective of such a discourse, which on the one hand illuminates and warms the technological contents, on the other hand expands them into an ethical and transcendent horizon through the wise expansion of the mentality of the researcher, called to become familiar with a new scientific approach enriched with contents.

The set of these five ethical relationships that now we resume as:

$$Data\ Ethics \rightarrow Poor\ Big\ Data \qquad (20)$$

$$Model\ Ethics \rightarrow Minority\ and\ Meekness \qquad (21)$$

$$Learning\ Ethics \rightarrow Personal\ Relationality \qquad (22)$$

$$Validation\ Ethics \rightarrow Pacification \qquad (23)$$

$$Presentation\ Ethics \rightarrow Purity\ of\ Hearth \qquad (24)$$

provides a paradigm that is not only exploratory but declarative of a process of re-reading technology in an ethical key, primarily involving the human person in the rediscovery of his dignity and his transcendence not available for reductionism.

Poverty, minority and meekness, relationality, pacification and purity of heart, are characteristics of the Service of Love emerging from an AI system designed within the Dynamical Techno-Algor-Ethical Composition. In fact, in these characteristics converge the ethical tensions typical of technology, such as power, justice and responsibility (Social Ethics of Technology), which are always lurking as risks and at the same time are always available as opportunities for growth along the entire path of a technology oriented towards the common good and fulfillment (Transcendental Motives of Technology). These features agree with the definition of Technological Sustainability and therefore define a sustainable Algor-ethics for AI.

We must also add that these relationships are by no means static, as we have been able to observe along their description, that is, they cannot be applied without the participation of the human person and, in some way, without his ethical 'conversion' prompted by the unexpected discovery of a treasure of depths and potentials, where instead only technical-ities appeared. The typical creativity of technological planning is precisely the sign and the call to such depths, to be discovered as long as we humbly accept a cross-disciplinary integration that adds to the knowledge and technical skills sapiential contents, which, as we wanted to show, are reasonably even if unexpectedly applicable in the context of a new extended rationality.

In this way, both the technological process and, above all, the researcher are involved in an attractive and challenging dynamism: poverty, conceived in its relationship with the totality of reality and with its complexity, stimulates the expression of human wealth and of the ordered creativity of the project; minority and meekness highlight the dual attitude of man to keep supervision and to maintain a principle of reality that can never be reduced to technique or simulation; the typical relationality of knowing and learning spurs the Person to keep alive the communal sense of acting, also scientific, constantly re-proposing the hunger and thirst for truth, sometimes forgotten; the pacification along the validation process serves to maintain the balance between the parts, with the wisdom of the rainbow, which relates distances, illuminating the colors of diversity, without compromises. Finally,

the purity of heart of the researcher allows the development of a technology and an AI capable of putting the relational dimension, purely human, in the foreground, both in the planning phase and in the implementation phase, recognizing the technological function of transparency and service with respect to the human person.

The difficulties in applying this approach, which is based, as we have seen, on a multi-dimensional and dynamic ethical composition, then becoming concrete in the various technological processes, allowing concrete ethical paths to emerge, mainly reside in the required change of mentality. On the one hand, the latter is in some way imposed by the conspicuous overcoming of mechanistic determinism; on the other, however, in the face of 'quantity' and 'speed' (two characteristic traits of the post-modern technological era) it risks being practically forgotten due to lack of time and therefore attention. It is a matter of a change of mentality that introduces an ethical horizon as well as a transcendent horizon, as not only necessary but also propulsive resources; who knows if a new scientific beatitude will not be born from this!

## 6. Conclusions

The uncertainty emerging from the deeper knowledge of the Cosmos in its complexity stimulates the human search for the truth, orienting it toward confidence instead of remaining in the fear caused by a relative and myopic gaze. This trust must also be strengthened in the context of technological development in which the available opportunities and results, so accelerated, can divert humanity from its dignity in favor of an imbalance towards technology.

In this article we have proposed a path to structure the foundations of a Technological Sustainability by building a founding ethical system. In a person-centered reference, the first step was to recognize the close and bidirectional link between technology and anthropology, based primarily on the results proposed by Benanti [1,3,4]. Subsequently, we have identified some specific pillars characterizing the ethical dimension of technology at the service of man, defining a horizontal dimensionality constituted by the Social Ethics of Technology (responsibility, justice and power) and a vertical dimensionality described by the Transcendental Motives of Technology (anthropology, freedom- creativity and eschatology). The purpose of this approach was to recognize, precisely in the link between technology and anthropology, on the one hand the most consistent ethical traits, and on the other the 'engines' that can push technology towards a wide-ranging development [5]. We then set out to compose these founding pillars in a dynamic interconnection, drawing on which it could be possible to develop a 'Good Algorithm' each time, therefore ethically connoted. In order to proceed in this direction, we had to introduce a multi-dimensionality that would act as a glue, as a connection and as a guarantee of elasticity for each of the fundamental pillars. In particular, the Service of Love or Power of Service characterizes the main cardan joint, and represents the 'semantic and pulsating glue', while the Fulfillment of man represents the essential orientation for the development of every technological process. The almost heartbeat of the Service of Love which collects (systole) responsibility, power and justice to nourish them and sends them (diastole) for the benefit of the common good, is therefore accompanied by the constant flow or, rather, the constant orientation, towards the fullness of the human person.

The result was to connect the technology, placed at the service of the human person, with ethical and dynamic components necessary to then structure specific algorithmic references, that is a 'Good Algorithm'. We called this interconnection the Dynamical Techno-Algor-Ethical Composition, thanks to which we were able to propose a definition of Technological Sustainability strictly connected to the founding ethical coordinates, as illustrated in our proposal.

Finally, equipped with this dynamical composition, we turned to the elaboration process of an AI system, as an application proposal for the development of a corresponding algorithm. By splitting the AI algorithmic process into five subsystems, in order to carry out a structural ethical reflection, we have identified, as a resonance of the ethical framework

previously proposed, five corresponding Algor-ethical traits which, on the one hand make the algorithm ethically sustainable, and on the other hand orient the human person to a full appreciation of his dignity and his relationships. This is to recall the importance of a procedure in which the human person might be in the center of each technological development, to compose Good Algorithms in order to proceed along a sustainable human progress, rightly evaluating and using technology itself as a real symbol of an authentic oriented history.

**Funding:** This research received no external funding.

**Institutional Review Board Statement:** Not applicable.

**Informed Consent Statement:** Not applicable.

**Data Availability Statement:** Not applicable.

**Conflicts of Interest:** The author declares no conflict of interest.

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
