# Peer review of "Technological Sustainability and Artificial Intelligence Algor-ethics"

_sustainability, doi:10.3390/su14063215_

Round 1
Reviewer 1 Report
The abstract itself contains more than five grammatical errors. The use of language is poor. Many keywords are also misspelled. eg: "build an Echosystem Framework". Also there is no logical flow to the proposal.
The authors have failed to logically express the relation between sustainability and ethics.
Reviewer 2 Report
The quality and topic of this paper are consistent with the journal Sustainability, but it still took me a while to read it and there are still a lot of problems in it:
First of all, for references, numerical symbols [1], [2], [3]... are required to be marked in the body part of the paper. Although you use (Benanti 2018a), (UN 1987, 54), (Benanti 2018b, 127-128) to mark, but it is not very convenient for readers to find the reference quickly.
Secondly, I hope you can mark the glance mentioned in line 129, 144 and so on with numbers (1), (2), (3)...
Then, “Conclusion” in the Line 581 need to be changed to “6. Conclusion”
The overall logic of the paper was good, and can see that the author is a native English speaker. But I still need to put forward a suggestion, I suggest the author improves the conclusion part, because although the paper overall is good, but the conclusion is like lack of something, hope the author in the conclusion part makes a summary of the whole paper.
The author of this paper is undoubtedly a theoretical researcher, especially the theory put forward in the fifth part is impressive, but I also suggest the author to improve this part. As a reader, reading this part is very interesting, but I feel that it stopped abruptly before I could enjoy it. So I suggest adding a “5.6” Discussion Part to do a further discussion.
Reviewer 3 Report
It becomes very important to deepen the theme of ethics and technologies. In an epoch like this in which it can be assumed that technologies are transparent from a functional point of view but strongly impacting from an ethical point of view.
In developing the work, it is suggested to set up an empirical research on "ethical" perception and on the ethical approach to technologies both for those who use them and for those who design them.
Round 2
Reviewer 1 Report
The motive of the authors is to discuss a possible link between the development of an Algor-ethics oriented towards new technologies and AI, and its sustainability according to the Person-centered reference.
But this motive is missing in the article.
The quality of English has improved but in the current form it is still poor.
